# Wildfire Detection Probability of MODIS Fire Products under the Constraint of Environmental Factors: A Study Based on Confirmed Ground Wildfire Records

**Lingxiao Ying [1,2], Zehao Shen [1,\*], Mingzheng Yang [1] and Shilong Piao [1]**

[1] Ministry-of-Education (MOE) Key Laboratory for Earth Surface Processes, Institute of Ecology, College of Urban & Environmental Sciences, Peking University, Yiheyuan Road 5, Beijing 100871, China; ylxcqwez@pku.edu.cn (L.Y.); ymz9907@pku.edu.cn (M.Y.); slpiao@pku.edu.cn (S.P.)

[2] Key Laboratory of Land Consolidation and Rehabilitation, Land Consolidation and Rehabilitation Center, Ministry of Natural Resources, Guanyingyuan West District 37, Beijing 100035, China

\* Correspondence: shzh@urban.pku.edu.cn; Tel.: +86-010-6275-1179

**Abstract:** The Moderate Resolution Imaging Spectroradiometer (MODIS) has been widely used for wildfire occurrence and distribution detecting and fire risk assessments. Compared with its commission error, the omission error of MODIS wildfire detection has been revealed as a much more challenging, unsolved issue, and ground-level environmental factors influencing the detection capacity are also variable. This study compared the multiple MODIS fire products and the records of ground wildfire investigations during December 2002–November 2015 in Yunnan Province, Southwest China, in an attempt to reveal the difference in the spatiotemporal patterns of regional wildfire detected by the two approaches, to estimate the omission error of MODIS fire products based on confirmed ground wildfire records, and to explore how instantaneous and local environmental factors influenced the wildfire detection probability of MODIS. The results indicated that across the province, the total number of wildfire events recorded by MODIS was at least twice as many as that in the ground records, while the wildfire distribution patterns revealed by the two approaches were inconsistent. For the 5145 confirmed ground records, however, only 11.10% of them could be detected using multiple MODIS fire products (i.e., MOD14A1, MYD14A1, and MCD64A1). Opposing trends during the studied period were found between the yearly occurrence of ground-based wildfire records and the corresponding proportion detected by MODIS. Moreover, the wildfire detection proportion by MODIS was 11.36% in forest, 9.58% in shrubs, and 5.56% in grassland, respectively. Random forest modeling suggested that fire size was a primary limiting factor for MODIS fire detecting capacity, where a small fire size could likely result in MODIS omission errors at a threshold of 1 ha, while MODIS had a 50% probability of detecting a wildfire whose size was at least 18 ha. Aside from fire size, the wildfire detection probability of MODIS was also markedly influenced by weather factors, especially the daily relative humidity and the daily wind speed, and the altitude of wildfire occurrence. Considering the environmental factors' contribution to the omission error in MODIS wildfire detection, we emphasized the importance of attention to the local conditions as well as ground inspection in practical wildfire monitoring and management and global wildfire simulations.

**Keywords:** wildfire detection capacity; MODIS fire products; ground records; environmental factors; Yunnan Province in China

## 1. Introduction

Wildfire is a critical disturbance and intrinsic natural process in various terrestrial ecosystems, and it closely interacts with global environmental changes as intensified by anthropogenic factors [1,2]. The impacts of wildfires on vegetation dynamics and biodiversity maintenance [3,4], air pollution and soil erosion [5,6], forest management and ecosystem services [7,8] are not trivial. For this reason, wildfires are routinely investigated in many countries, with most wildfire studies traditionally relying on ground records accumulated by forestry departments and institutions [9,10]. Alternatively, remote sensing data are now increasingly used for recording wildfires and their environmental impacts at regional to global scales [11,12]. In particular, the satellite-based Moderate Resolution Imaging Spectroradiometer (MODIS) offers advantages over other data sources for studying the occurrence and extent of wildfires [13,14] by providing two key fire products: active fire detections and burned area estimates. The former records wildfire hotspot locations while the latter quantifies the burned areas. Both are widely used as data sources in many large-scale analyses of wildfire activity and environmental impacts, climate change scenario simulations, and vegetation response projections [2,15]. Meanwhile, MODIS is also commonly used in monitoring wildfire events in regional forest management [16,17].

Such widespread applications of MODIS in wildfire detection and management, however, are necessarily accompanied by errors originating from inherent limitations of the data source including its spectral properties and the spatial and temporal resolutions [18,19]. On one hand, MODIS detection errors for the burned area have been controlled reasonably well, and its determinants (e.g., land use type) have been increasingly discussed, mainly based on multi-remote sensing data [20–22]; on the other hand, for wildfire occurrence detection, as widely validated with other remote sensing data, the ratio of the commission error is much lower than that of the omission error in the MODIS products. For example, Hantson et al. [23] and Schroeder et al. [24] respectively compared Landsat images and ASTER fire products to MODIS active fire products (Collection 5) for detecting regional wildfire occurrence, and both studies found low commission error ratios of about 2%. In contrast, how to deal with the omission error of MODIS active fire products is still an issue to be resolved, although it is considered as mainly related to fire size [23,24]. Recently, the latest products of MODIS wildfire occurrence detection (Collection 6) were released with reasonable improvements that include better commission error control, despite the variation across different regions [25]. Omission error, not surprisingly, has not improved that much, especially for areas such as Southeast Asia, where the wildfire detection probability was lower than 7%, while that of Central Asia was less than 12.5%. Additionally, these products generally showed a low detection capability for small-sized wildfires [25]. Although various types of remote sensing data have been compared to cross-validate their wildfire detection capacity, no consensus has yet been reached [26–28]. Hence, it is necessary to consider and apply other data sources such as the wealth of archived ground records to reasonably assess the probability and accuracy of MODIS wildfire occurrence detection as well as the determinants of errors.

Comparisons of MODIS and ground-based data for error cross-validation have been pursued for a variety of research topics such as aerosol optical depth [29], land surface albedo [30], plant phenology [31], and snow cover [32], to name a few. A comparison between these two approaches with respect to regional burned area has also been recently undertaken. For instance, Mangeon et al. [33] compared MODIS and ground-based data from 26 wildfires across North America and found that differences between the two sources increased as the temporal resolution was enhanced. Li et al. [34] compared MCD45 products for burned areas with ground forest fire survey data summarized at the provincial level in China and found that estimates derived from both data sources were comparable across the country, yet notable differences persisted among provinces. It is worth mentioning that Fusco et al. [35] first studied the detection rate of MODIS fire products with very large numbers of ground-based samples, and suggested that annual cloud cover and leaf area index in the representative year (2007 in their study) were significant predictors of MODIS wildfire detection in the USA.

Robust comparisons of wildfire records between MODIS and ground-based data are still scarce. This leaves the accurate assessment of MODIS data in wildfire detection capacity, or omission error, a key research aim, albeit challenging to achieve.

In this study, by comparing to the determined ground records of wildfire events, we focused on the omission error of MODIS fire products covering Yunnan Province in Southwest China, which is intensively impacted by drought-related wildfires in its ecosystems [36], representative of the environment under the prominent influence of the Indian Ocean Monsoon (IOM) and the Pacific Ocean Monsoon (POM). Considering that monitoring wildfire occurrence will prefer using remote sensing sources to human patrols for regional management practices, we aimed to estimate the true wildfire detection capacity of MODIS in a heterogeneous environment, and unlike Fusco et al. [35], we quantified the contribution of instantaneous and local environmental factors such as weather conditions to the omission error in multiple MODIS fire products. We believe that a reasonable estimate based on the comparison and environmental impact analysis will be helpful for understanding such error sources and is crucial for applications of MODIS in forest fire management.

## 2. Materials and Methods

### 2.1. Study Area

Yunnan Province is located in Southwest China (ca. 21°08′–29°15′N and 97°31′–106°11′E), adjacent to the Indo-China Peninsula (Figure 1). Topography in Yunnan Province is mainly composed of a discontinuous plateau in the central and east with an elevation range of 1500–2500 m a.s.l., and characterized by longitudinal mountain ranges and gorges mainly on the western part. Across the province, the range of mean annual temperature is about 5.91–23.90 °C, while that of annual precipitation is 563.90–2452.21 mm approximately, according to the historical observations of meteorological stations [37]. Importantly, Yunnan is home to one half of native plant species in China, with only 5% of national land area [38]. The climate in Yunnan is jointly influenced by the IOM from the southwest, and the POM from the east. The two monsoons thus bring complex weather dynamics, especially that of precipitation seasonality [39,40]. The prominent topographic relief amplifies the regional climatic variation, represented by an obvious dry season typically starting from December to the following May [41]. Concerned by global climate change scenarios, wildfires are now widely monitored across China as well as in the Indo-China Peninsula and its surrounding regions, which are dominated by the IOM and POM [1,42]. The weak capacity of MODIS fire products to detect wildfire occurrence in this region, as clearly shown by Giglio et al. [25], therefore strengthens the impetus for estimating the omission error in the data using confirmed ground records, and for exploring the influence of potential factors on this margin of error.

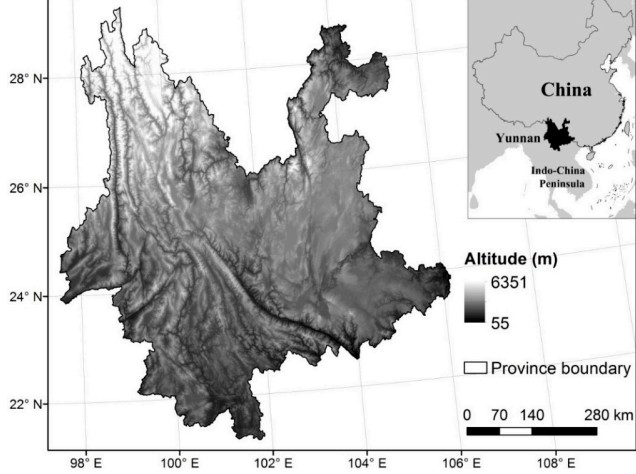

**Figure 1.** The location of Yunnan Province in China and its elevational variation pattern.

## 2.2. Wildfire Data Sources

The ground-based wildfire data was obtained from the Fire Prevention Office of the Yunnan Forestry Department. To ensure the uniformity and integrity of this data, we only used those records from December 2002 to November 2015 (i.e., 2003–2015); this included detailed information such as occurrence time and duration, location, fire size, reason of burning, major vegetation type burned, and estimated economic loss for each of the 5145 wildfire events recorded. All wildfire locations were geo-referenced as a wildfire occurrence point layer, by using the ArcGIS v.10.3 platform, in which we applied the Asia North Albers Equal Area Conic projection.

For the comparison with the ground-based data, we used three types of MODIS data on fires (Collection 6) in the same period: two active fire products, MOD14A1 and MYD14A1, and the burned area product MCD64A1 from NASA (U.S. National Aeronautics and Space Administration) [43]. As the temporal resolution is 1 day, the spatial resolution of MOD14A1 and MYD14A1 is 926.62 m for daily fire locations (grid cell), with low, middle, and high confidences for fire detection. We merged both active fire products by designating those grids with the same location and adjoining dates as one wildfire event; correspondingly, adjacent grids of fire with the same date were necessarily merged into one event. Thus, wildfire polygon vector layers with high, middle, and low confidences were detected and could be shown using ArcGIS v.10.3. Meanwhile, the center of each wildfire polygon was extracted as the wildfire occurrence point for data analyses [44]. Likewise, the MCD64A1 product, which has a spatial resolution of 463.31 m and a temporal resolution of eight days, was treated following the same procedure to obtain a more reasonable estimate of independent fire events.

## 2.3. Environmental Factors

The global digital elevation model, SRTM v.4 [45], was used to extract key topographic information associated with wildfire events. This model has a spatial resolution of 3″, which is approximately 90 m near the equator [46]. Both the altitude and slope angle of each wildfire event were extracted and calculated in the platform ArcGIS v.10.3, while the local slope position (valley bottom, lower slope, middle slope, higher slope, mountain top, and flat area) of each confirmed ground wildfire event has already been provided in their records, according to the major part of the fire location.

As weather conditions are recognized as a prominent error source in wildfire remote sensing data [47], we used the daily mean temperature, precipitation, relative humidity, and wind speed for their associations with ground wildfire records. This weather data were obtained from the National Meteorological Information Center of China [37] and included the records from 36 meteorological stations in Yunnan Province during the study period. For each variable, a thin-plate smoothing splines (TPSS) interpolation model was applied to the weather records of 36 geo-referenced data points (i.e., meteorological stations) with their respective altitude used as a covariate [48] to generate a daily weather map across the entire study area. The TPSS interpolation model with the altitude as covariate was shown to be superior to other interpolation methods [49,50], especially for the mountainous and hilly regions in China [51,52]. These interpolations were performed in ANUSPLIN v.4.3 software by applying a generalized cross-validation to obtain the best simulation results [53]. To test their accuracy, we compared a total of 1000 random samples from all interpolation simulations with factual records of the corresponding meteorological stations. The results showed high accuracy for each variable involved as the $R^2$ of the TPSS models for the daily mean values of temperature, precipitation, relative humidity, and wind speed were 92.65%, 96.58%, 86.40%, and 84.62%, respectively.

Vegetation information of each ground-recorded wildfire event included vegetation type (forest, shrubs, or grassland), stand age class (young, middle-aged, near-mature, mature, or over-mature), and leaf area index (LAI). The former two were taken directly from existing ground records, while the vector data (at a map scale of 1:100,000) of land use/cover type in Yunnan Province in the year 2000 from the National Earth System Science Data Center of China (NESSDCC) [54] were also used to identify the vegetation type of high-confidence wildfire events in MODIS active fire products for the following analysis. The LAI data was extracted from the MODIS product MO/YD15A2H at a spatial

resolution of 463.31 m. For each wildfire event recorded, considering the quality of LAI products, the 8-day composite LAI value with a higher quality when the wildfire occurred was applied at its location (i.e., grid cell). Values for those grid cells lacking vegetation cover, namely construction areas, bare land, water bodies, and snow-covered parts, were all set to zero in the LAI dataset.

Human activities could be considered as potential influential factors of MODIS wildfire detection [35]. Vector data of the human residences and roads of all levels in Yunnan Province for study period were downloaded from NESSDCC (at a map scale of 1:250,000) [54]. Then, for each wildfire event, the distances to the nearest village-level residence and road were regarded as influential factors and calculated in ArcGIS v.10.3.

## 2.4. Statistical Analysis

We compared the temporal wildfire patterns of ground records to MODIS active fire products with high confidence within a given fire season (from December to following May) and non-fire seasons (from June to November). For ground records and the high-confidence records of MODIS active fire products, the kernel density estimation (KDE) algorithm was applied to the wildfire occurrence point layer in ArcGIS v.10.3 to visualize the spatial aggregation characteristics of wildfires across Yunnan, and to identify the locations with higher wildfire occurrences relative to neighboring areas [55,56]. Moreover, we also compared the number of wildfire events of these two approaches among the vegetation types.

To estimate the omission error of MODIS fire products, we searched the neighborhood of each ground-recorded wildfire point, referring to the time of occurrence confirmed by ground records. Active fire products of low, middle, and high confidences, and burned area product here were all taken into account to improve the wildfire detection probability of MODIS [57,58]. Thus, in ArcGIS v.10.3, we considered the case where there were no fire records (empty grid cell) of MOD14A1 and MYD14A1 (with temporal difference no more than one day) overlapping a neighborhood buffer of ground-recorded wildfire point as the searching radius of the buffer equaled the spatial resolution of the corresponding MODIS fire products. We also took into account the case of no fire records derived from MCD64A1 (with temporal difference no more than eight days) overlapping the corresponding neighborhood buffer. Of note, the sizes of the buffer encompassed more than 95% of the ground-recorded wildfire events based on the reported fire size, and therefore for these cases above-mentioned, we must logically conclude that the omission occurred with regard to the corresponding ground record [35].

For statistical analysis in this study, the confirmed ground records detected by MODIS were scored as "1" and those omitted as "0" samples. Considering the relative low probability of MODIS wildfire detection, we expected to encounter many "0" samples in our study area. In general, with imbalanced samples like these, a classifier model algorithm can improve the accuracy of the majority class, but ignores that of the minority class to reach the minimum classification error ratio (CER). Thus to ensure the classification accuracy of both "1" and "0" samples, we performed bootstrap-sampling with 1000 iterations and sampled the omitted ground records to compose the "1" and "0" samples with a ratio of 1:1.5 in each iteration. Then, a random forest (RF) model was used to interpret the variation in the wildfire detection probability of MODIS, with twelve influential variables included (Table 1). The best combination of variables in the model (i.e., with minimum generalization error) was detected using the criterion of minimum CER of the out-of-bag (OOB) data, which can specifically convey the accuracy of the RF model [59]. In this method, with 1000 "trees" to grow in a RF model comprising each variable combination, about one-third of all samples, our so-called OOB data, were left unused to serve as the training data for each tree. Then, the vote, classified as "1" or "0", of each OOB sample was detected among the trees not containing the sample as a training case, but rather as a testing case. Thus, when compared with the observations, the CER of the OOB data of the model could consequently be calculated. Furthermore, the relative importance of each predictor variable in the RF model could be expressed by an increase in the CER (or accuracy decrease) of the OOB data when the value of the variable was permuted randomly [60]. Finally, partial dependence plots were

used to show the marginal response of MODIS wildfire detection probability to each variable [59,61]. For each variable, the *x*-axis of the partial dependence plot was within the value range of the variable, and for a specific value of the variable (*x*-axis value), the corresponding *y*-axis value of the partial dependence plot (i.e., MODIS wildfire detection probability) could be calculated by the best RF model. Practically, the variable was fixed with the specific *x*-axis value, and the predictive was averaged as the related *y*-axis value with all combinations of other variables in the model. These statistical analyses were all performed on the R platform v3.4.1.

**Table 1.** The influential factors for estimating the wildfire detection probability of MODIS, based on the ground records of the wildfire event.

| Factor | Variable (units) | Average (±Standard Deviation) | Descriptions |
|---|---|---|---|
| | (ln-) Fire size (ha) | 1.96 (±1.73) | Natural logarithm of fire size used due to most samples with small fire size (also referring to $p = 0.6413$ based on the Kolmogorov–Smirnov test for null hypothesis of lognormal distribution) |
| Weather | Temperature (°C) | 17.80 (±4.91) | Daily mean temperature |
| | Precipitation (mm) | 0.24 (±1.01) | Daily precipitation |
| | Relative humidity | 0.47 (±0.12) | Daily mean relative humidity, 0~1 |
| | Wind speed (m/s) | 2.48 (±0.88) | Daily mean wind speed |
| Vegetation | Vegetation type | (Categorical) | Forest, shrubs, and grassland |
| | Stand age class | (Categorical) | Young, middle-aged, near-mature, mature, and over-mature |
| | LAI | 1.10 (±1.04) | Leaf area index when wildfire occurred |
| Topography | Altitude (m) | 1738.41 (±506.78) | |
| | Slope (°) | 12.79 (±9.15) | |
| | Slope position | (Categorical) | Bottom, low, middle, high, top, and flat |
| Human activity | Residence distance (m) | 492.77 (±438.95) | Distances to the nearest rural residence |
| | Road distance (m) | 776.15 (±1315.99) | Distances to the nearest rural road |

## 3. Results

### 3.1. Temporal and Spatial Patterns in Wildfire Detection Differences

During the 13-year study period, the annual number of high-confidence wildfire records derived from the MOD14A1 and MYD14A1 products averaged 1097.08 events/year, or 2.77 times in average the number of ground-recorded wildfires across Yunnan Province (Figure 2). Both MODIS and the ground-based data agreed that the majority of records occurred within the fire season from December to the following May, corresponding to 96.78 ± 1.84% for the former and 98.73 ± 2.11% for the latter during the study period (Figure 2). For the fire season, there was a significantly decreasing trend of the annual ground-recorded wildfire events ($p = 0.0016$), but there was no significance for the MODIS-based data ($p = 0.5530$) (Figure 2).

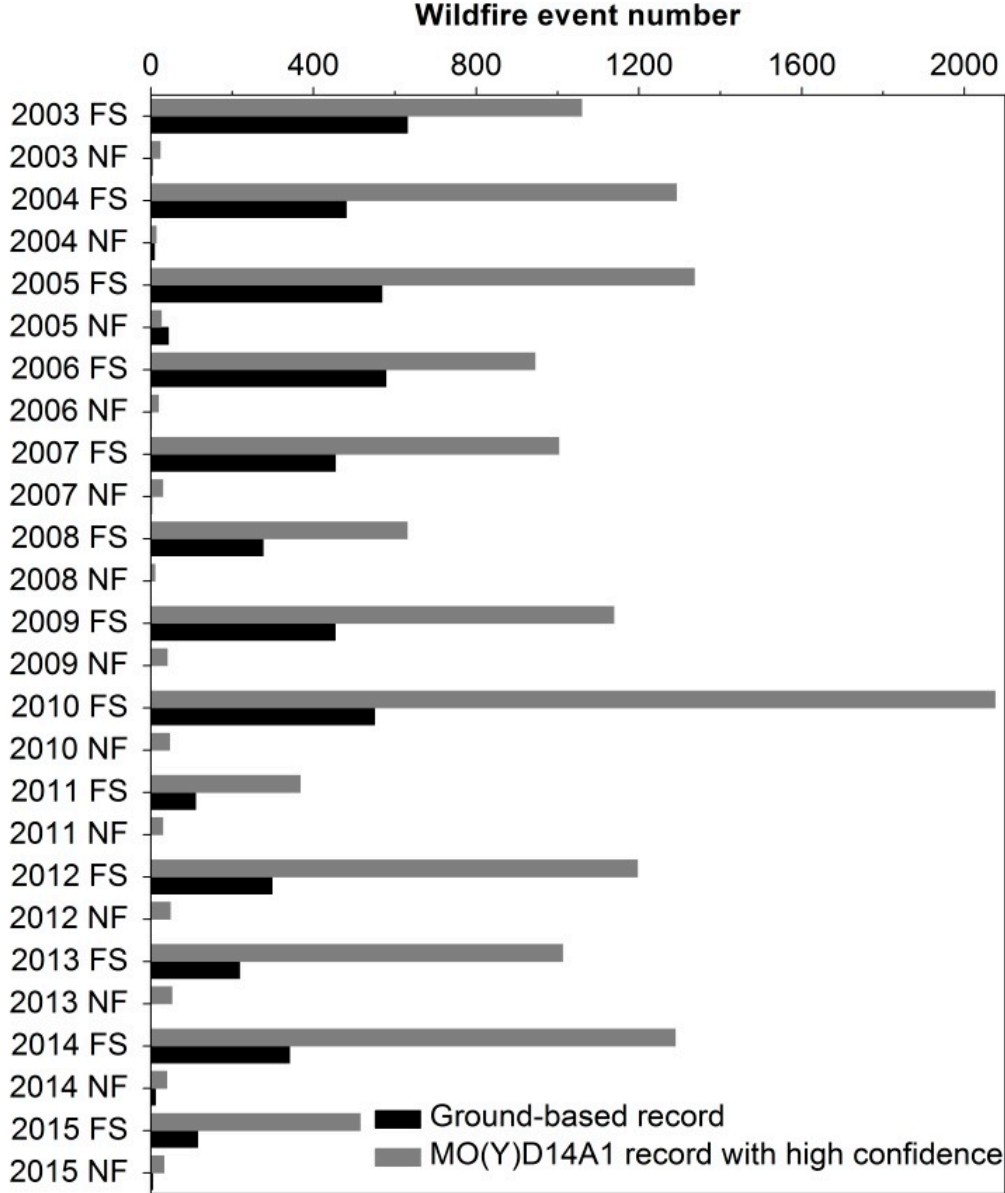

**Figure 2.** The inter-annual changes (2003–2015) of the number of wildfire events based on ground records and the high-confidence records of MOD14A1 and MYD14A1 (MO(Y)D14A1), with the distinction of fire season (FS) and non-fire seasons (NF) in the study area.

The KDE results indicated spatially inconsistent wildfire distributions between two data sources. Based on the ground records, there were several obvious aggregation regions of wildfire event in the study area such as the Three Parallel Rivers (TPR) region of northwestern Yunnan (Figure 3a), while a much higher wildfire density was identified by the high-confidence records of MODIS active fire products aggregated in the TPR region (Figure 3b). Meanwhile, based on MODIS, relatively high wildfire density was also occupied in southern Yunnan, especially the lowland areas of Sipsongpanna and Jinggu County (Figure 3b). However, the MODIS products must not always be effective for recording wildfires (Figure 4). Although in most areas of the TPR region and the south, the MODIS products detected more wildfire events than the ground records, in vast areas of central and northeastern Yunnan, the wildfire events recorded by MODIS were less than those ground-recorded, even less than half of the ground records (Figure 4).

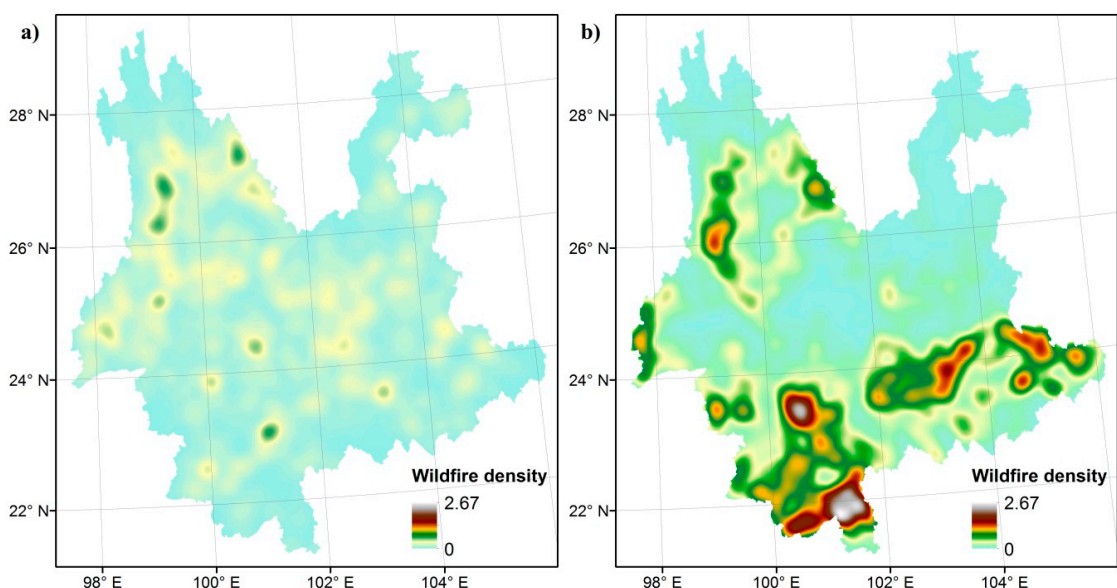

**Figure 3.** The spatial patterns of wildfires. (**a**) Kernel density estimation (KDE) of the number of wildfire events based on ground records, and (**b**) KDE of number of wildfire events based on the high-confidence records of MOD14A1 and MYD14A1 (unit: $10^{-3} \cdot ha^{-1}$).

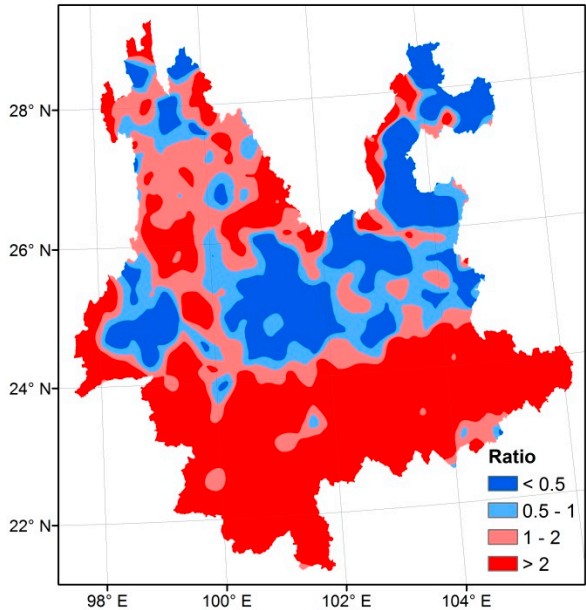

**Figure 4.** From the KDE results, the ratio of number of wildfire events based on the high-confidence records of MOD14A1 and MYD14A1 to that based on the ground records.

### 3.2. Wildfire Detection Capacity of Multiple MODIS Fire Products

Only 571 out of the 5145 ground observed wildfire events were detected and included by multiple MODIS fire products, which corresponded to a nearly 90% omission. The mean wildfire detection proportion was 11.14% and 8.22% in the fire and non-fire season, respectively (Figure 5a,b). For the fire season, a significant trend of increasing wildfire detection proportion was indicated across years ($p < 0.001$), even though a significant opposite trend was detected for the number of wildfires recorded with certainty ($p = 0.0020$) (Figure 5a). The wildfire events detected by multiple MODIS fire products were mainly located in the central Yunnan Plateau and the areas of longitudinal mountain ranges and gorges in the northwest of Yunnan, with numerous omissions in the northeast and southwest (Figure 5c).

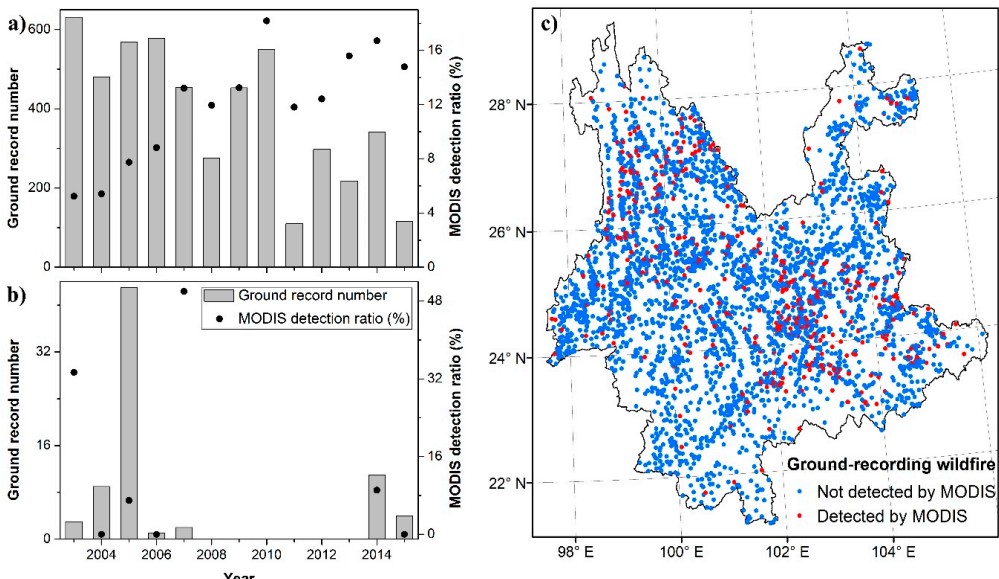

**Figure 5.** The wildfire detection capacity of multiple MODIS fire products. The inter-annual variation of confirmed ground wildfire records and their detection proportions by MODIS are shown within the (**a**) fire season and (**b**) non-fire season across the study area. The spatial distribution of confirmed ground-recording wildfires detected and omitted by MODIS are shown in (**c**).

Additionally, when considering the vegetation type burned, despite the total number of wildfires recorded by MODIS being far greater than that of the ground records among all types, the wildfire detection proportions by MODIS were only 11.36% in forest, 9.58% in shrubs, and 5.56% in grassland, respectively (Figure 6). Furthermore, with regard to some common tree species in the forest type in Yunnan Province, the wildfire detection proportion was 12.22% for *Pinus yunnanensis* (166 detected in 1358 cases); 13.10% for *Cunninghamia lanceolata* (49 detected in 374 cases); 12.96% for *Quercus* spp. (35 detected in 270 cases); 5.67% for *Pinus kesiya var. langbianensis* (14 detected in 247 cases); 9.52% for *Pinus armandii* (16 detected in 168 cases), and so forth.

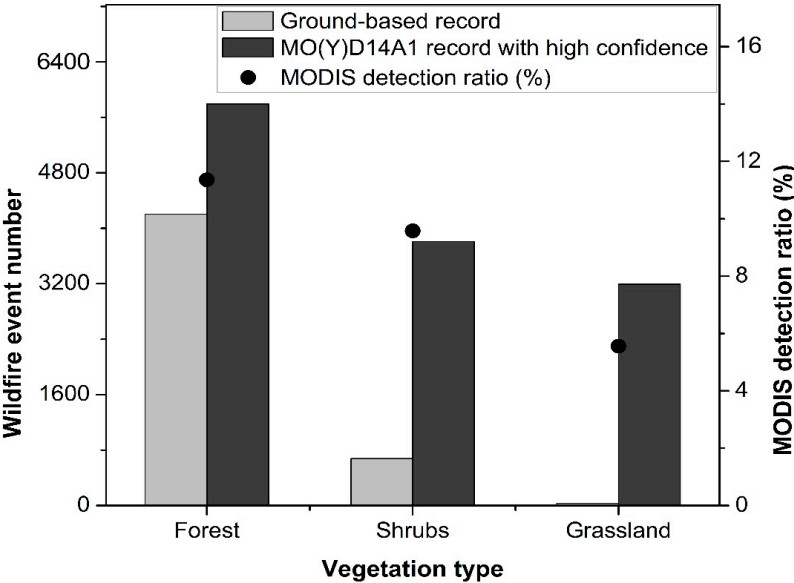

**Figure 6.** The wildfire detection ratio of multiple MODIS fire products varied with vegetation type, while the differences in the wildfire number between the ground records and the high-confidence records of MOD14A1 and MYD14A1 (MO(Y)D14A1) are also shown.

### 3.3. Determinants of Wildfire Detection Probability

For the best random forest (RF) model of wildfire detection probability by MODIS, the mean CER based on OOB data was $30.69 \pm 1.33\%$ for 1000 iterations, a value significantly different from the randomly predicted error proportion of 50% ($p < 0.001$). Four variables entered the best model more than 900 times. Three of these, (ln-) fire size, relative humidity, and wind speed, were retained in all the best RF models, while altitude was respectively selected in 940 of 1000 iterations (Figure 7). In terms of the relative importance, the top variables were (ln-) fire size, daily relative humidity, daily wind speed, and altitude. Daily temperature and precipitation were other important weather factors, as the two variables were selected more than 500 times among 1000 iterations. In contrast, vegetation type and structure, represented by stand age class and LAI, made little contribution to the wildfire detection probability (Figure 7). Moreover, local scale topographic features such as slope angle and slope position showed little impact on the MODIS detection ratio, and nor did the potential of human activity represented by residence distance and road distance reveal prominent impact (Figure 7).

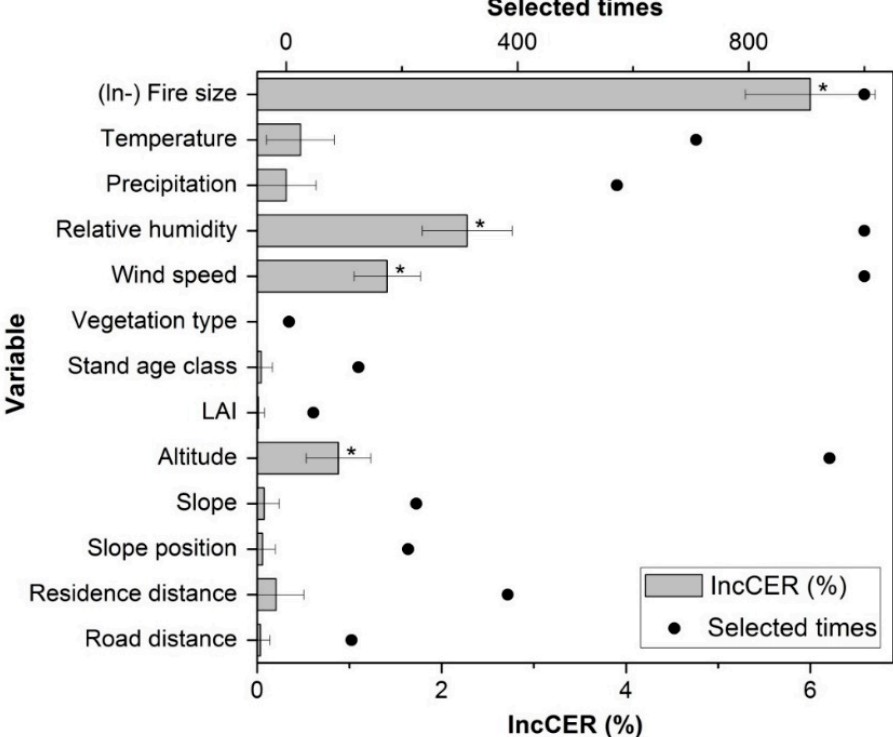

**Figure 7.** The times of selection (within the best random forest (RF) model) and mean (with standard deviation) relative importance of each predictive variable among 1000 iterations with bootstrapping sampling. Note that the relative importance was reflected by the increase in the classification error ratio (IncCER) of 'out-of-bag' data when the value of the variable was permuted randomly in the RF model, and * represents the selected times over 900.

With fire size increasing, the marginal wildfire detection probability by MODIS showed a three-stage response (Figure 8a). The fire size of 1 ha (i.e., (ln-) fire size equaling 0), acted as a threshold, below which the detection probability stayed as low as 15%, then it grew almost linearly with increasing fire size until another threshold value of about 148 ha ((ln-) fire size = 5), above which the detection probability stabilized at about 60% (Figure 8a). The detection probability showed an opposite response curve to the relative humidity (RH), declining from 55% at RH $\leq$ 0.2 to 30% at RH $\geq$ 0.7 (Figure 8b). The detection probability showed a more abrupt increasing response to the wind speed at a threshold of about 2 m/s (Figure 8c). Moreover, the omission error of multiple MODIS fire products revealed a complex response to altitude (Figure 8d), although with a gentle increase of probability

from 1000 m to 2500 m a.s.l.. Other variables only had a limited influence on the detection capacity (Figure A1).

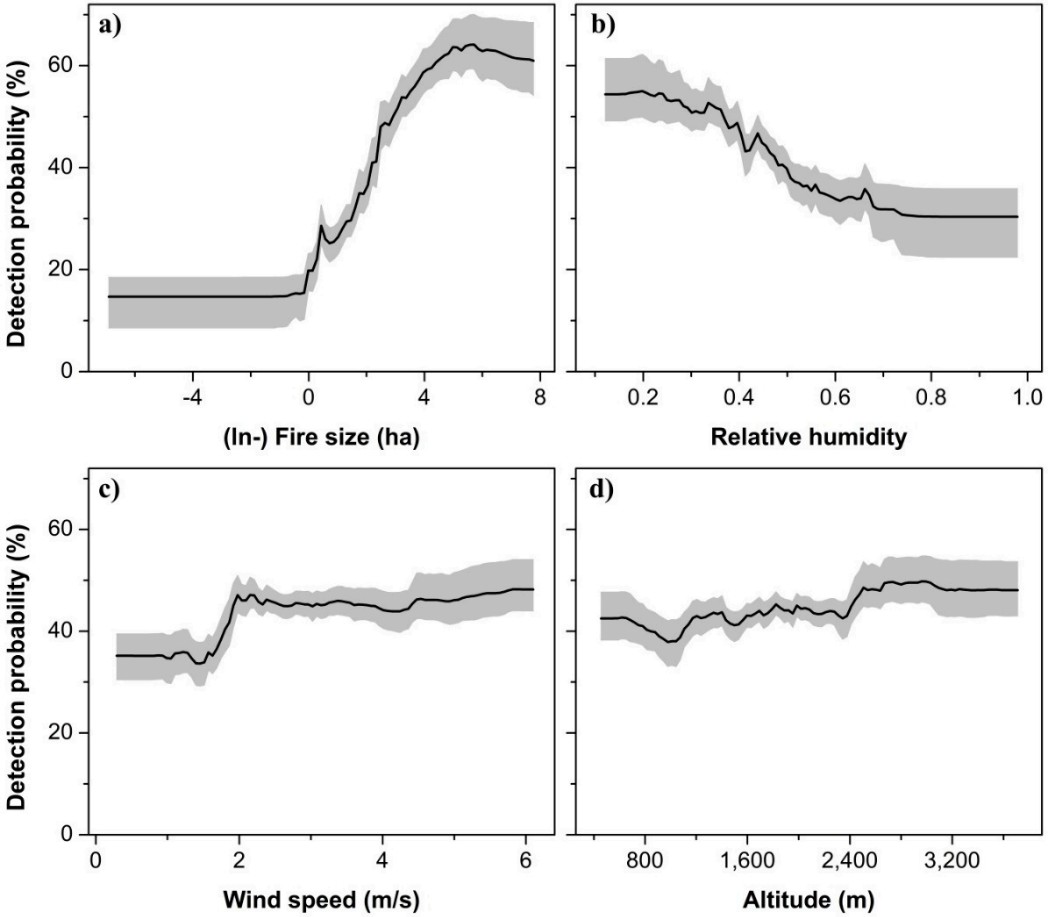

**Figure 8.** The marginal responses of MODIS wildfire detection probability to the four most important variables, with the decrement of the variable's relative importance from (**a**) to (**d**) in order, while grey parts represent the 95% confidence interval.

## 4. Discussion

Across our study area, the MODIS fire products recorded far more wildfire events than the ground monitoring records (Figures 2 and 3a,b). According to the wildfire survey procedures of the Fire Prevention Office in the Forestry Ministry of China, many small fires that did not cause significant loss to forests or social properties, especially those extinguished without requiring department response, were not included in records, such as the prescribed burnings for pastoral maintenance or artificial forest plantations, straw burning for fertilizers as well as recreational fires for picnics, with the same practices in the corresponding departments of many countries [17,35]. Meanwhile, the ground records may have omissions which generally result from wildfires that occurred in some remote areas [34,35]. Not to be overlooked, ground objects with high temperatures such as chimneys and tin roofs in cities and towns could induce further commission errors by MODIS [23,62].

However, for those wildfire events actively recorded and confirmed in wildfire management, the MODIS omission error was substantial, despite applying all active fire and burned area products, which thus confirmed the relatively low detection probability of the Collection-6 products of MODIS for the study region [25]. The size of the wildfire event was the most crucial factor influencing the detection probability, since small-sized wildfires are often omitted in MODIS fire products; 13.66% confirmed wildfires had a burned area smaller than 1 ha, and 69.85% of those had a burned area smaller

than 18 ha, which could approximate the minimum fire size that made the wildfire detection probability of MODIS reach 50% in Yunnan Province (Figure 8a). Fusco et al. [35] indicated that MODIS had a 50% probability of detecting a wildfire that grew to at least 10 ha in eastern USA, and several studies have also suggested that the critical values of fire size that sensitively affected the detection capacity of MODIS varied among regions with different vegetation types [63,64], indicating a prominent regional difference that requires further effort to understand the underlying mechanisms with some other influential factors.

Efforts have long been invested in exploring how environmental factors may affect wildfire occurrences, based on either MODIS data [12,15,16] or ground monitoring records [65,66]. This study suggests that instantaneous and local factors could also influence the MODIS wildfire detection capacity to varying extents (Figure 7). For example, dry weather conditions could promote MODIS wildfire detection [47], based on the thermodynamic principle that the moister the air, the more it absorbs energy, thus reducing wildfire radiant power detected by MODIS. Moreover, the approximately logistic negative correlation was revealed in the relationship of MODIS wildfire detection probability and relative humidity (Figure 8b). Although common sense would suggest that wildfire occurrence is also negatively associated with relative humidity, the small number of wildfires may not necessarily be accompanied with a low detection probability (Figure 5a). In addition, Peterson and Wang [47] suggested that a certain wind speed may remarkably enhance the wildfire radiative power flux, thus making it easier for MODIS to detect, as also indicated, but not very prominent in our results (Figure 8c). Meanwhile, a high wind speed may also accelerate the burning rate, and result in a large-size wildfire to increase MODIS detection probability. Altitude was another important factor related to omission error in MODIS wildfire detection, but its complex response may be due to the complex topographic change along the altitudinal gradient in Yunnan Province (Figure 8d), as it has been suggested that topographic roughness could affect MODIS wildfire detection [67].

When estimating fire regime using MODIS fire products, Chen et al. [17] suggested that the detailed field data were difficult to obtain in China, and that official statistics typically underreport fire events for management and technical reasons. This study indicated a considerable omission error of MODIS based on confirmed ground wildfire records. On the one hand, it recommends more considerations when using MODIS data in regional wildfire monitoring not only in the study area, but also beyond. It highlights the necessity of the serious attention to local environmental conditions as well as the cooperation with ground inspection and patrols for instantaneous wildfire assessment and design of fire prevention policies and measures in a region. Additionally, the MODIS fire products are widely used as fire-location inputs for various dynamic global vegetation models (DGVMs) and general circulation models (GCMs) while the fire related uncertainties have already been discussed [68,69]. For example, Veira et al. [69] applied MOD14A1 data to the GCM ECHAM6-HAM2 to simulate global patterns in the wildfire-caused emissions, and found that important omission biases may be introduced by the uncertainties in MODIS fire products. Variation in one or more instantaneous and local environmental factors could also magnify and compound errors in MODIS wildfire products and propagate related uncertainty into any derived results. Thus, on the other hand, with respect to the omission error, we suggest that considering the local conditions such as relative humidity and wind speed as revealed in this study could also be feasible to calibrate the global wildfire simulations, although the limitations of resolution and cloud cover in MODIS fire products has long been recognized [69,70].

## 5. Conclusions

Wildfire occurrences recorded at the regional scale by MODIS fire products were more than twice that based on the available ground records of Yunnan Province in China, but the spatial patterns of wildfire density based on these two approaches had limited matches. For 5145 confirmed wildfire events of ground records in Yunnan within 2003–2015, only 11.10% could be detected by the multiple MODIS fire products (i.e., MOD14A1, MYD14A1, and MCD64A1). The omission error was primarily linked

with fire size, while weather conditions also eminently contributed to the wildfire detection probability of MODIS, especially the daily relative humidity and daily wind speed. These results highlight the importance of combining multiple sources of wildfire information and relevant environmental variables for the monitoring and risk assessment of wildfires in regional forest management and global wildfire simulations.

**Author Contributions:** All authors have read and agreed to the published version of the manuscript. Conceptualization, Z.S., L.Y. and S.P.; Methodology, L.Y., Z.S. and M.Y.; Formal analysis, L.Y., Z.S. and M.Y.; Writing—original draft preparation, L.Y. and Z.S.; Writing—review and editing, Z.S. and L.Y.; Funding acquisition, Z.S.

**Funding:** This research was funded by a project of the National Natural Science Foundation of China, grant numbers 41971228 and 41371190, and the Key Research and Development Plan of the Ministry of Science and Technology of China, grant number 2017YFC0505200.

**Acknowledgments:** We are grateful to Jun Wang (Land Consolidation and Rehabilitation Center, Ministry of Natural Resources, Beijing) for advising on this work.

**Conflicts of Interest:** The authors declare no conflict of interest.

## Appendix A

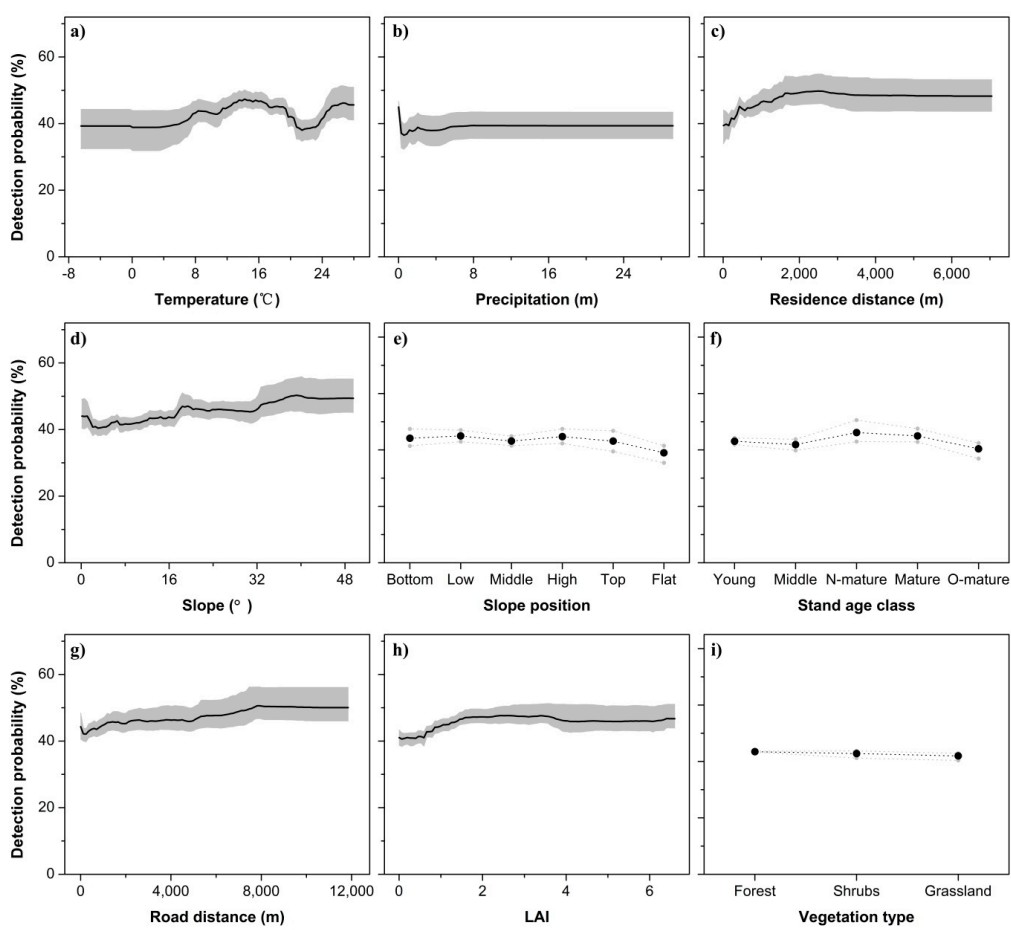

**Figure A1.** The marginal responses of MODIS wildfire detection probability to another nine environmental variables (with regard to those four variables in Figure 8), with the decrement of the variable's relative importance from (**a**) to (**i**) in order, while grey parts represent the 95% confidence interval.

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
