# Peer review of "Wildfire Detection Probability of MODIS Fire Products under the Constraint of Environmental Factors: A Study Based on Confirmed Ground Wildfire Records"

_remotesensing, doi:10.3390/rs11243031_

Round 1
Reviewer 1 Report
The study shows that there is inconsistent detection of fires via MODIS and ground-based records—many fires were detected by MODIS that did not occur in ground based records and at the same time, only a small percentage of ground based records were detected by MODIS. However, the percentage of ground based records detected is a well-defined statistic for the study. Although the study does not make progress in increasing the detection probability of fires using MODIS, the random forest analyses showed that certain factors limit the ability of MODIS to detect fires, such as fire size, and weather conditions on the day of a given fire. Thus, one of the main contributions of the study is to suggest avenues of future research to help solve problems with MODIS fire detection.
Here are some revisions
Section 2.1. Can you provide the range of mean annual temperature and precipitation across the province?
Line 170. What is temporary leaf area index? The word ‘temporary’ in this context is confusing—either drop it or explain.
Figure 5c should say ‘Not detected by MODIS’ rather than ‘No detected by MODIS’
Line 309, should say ‘limited influence’ rather than ‘few influence’?
Reviewer 2 Report
Ying et al. used 13 years of fire data comparing records of ground-based fire locations and three MODIS fire products in the Yunnan Province of Southwest China. Based on the ground-based records, the authors conclude that MODIS fire products have high omission errors. By modeling potential factors influencing MODIS fire detections authors point out four main factors contributing to MODIS fire detection capability in their study area.
Major comments
How complete was the ground monitoring fire records? I suspect that the ground monitored fire data has higher omission errors, especially in locations that are difficult to reach. Provide some information on how the ground fire records were acquired. How rigorous and comprehensive was the ground monitoring strategy and how reliable is the completeness of this data to be used as a “ground truth”? In relation to above comment: Was it possible for the Yunnan Forestry Department to map every fire that ever occurred? Considering the complex topography in parts of the study area, how well does the ground monitoring cover such locations (steep slopes, rugged topography etc).MODIS had a much higher fire detections than the ground records but only few of the MODIS detections matched the ground records. Is this mismatch not related to spatial and temporal mis-alignment inherent in your approach to compare these two product sources?
Minor comments
Additional description of basic characteristics of the MODIS fire products (such as temporal resolution etc) will be helpful to readers who are not familiar with this product.On line #267 – 269 authors mentioned distribution of MODIS fire detections by vegetation type. Presentation of a similar analysis for the ground fire records will be helpful. I suspect ground-based fire records miss a lot of fires in terrains and vegetation types that are difficult to reach.
Figure 5: Are the MODIS detections on the y-axis ratios or percentages?
Authors should consider cutting down the number of references (about 30%?) to include only the most relevant ones; 100 references seem too much for a paper of this nature.
Round 2
Reviewer 2 Report
Author’s need to improve on the writing in a way that clearly communicates their salient findings to readers. For instance, certain statements in the abstract are confusing, making it hard to understand without reading the entire manuscript. On line # 25, authors indicated that MODIS detected more fires than the ground data (at least twice as that detected by ground data). Then on line # 27 authors make a seemingly contrary statement indicating that “Only 11.10% of the 5145 ground-recorded wildfires could be detected using multiple MODIS fire products”. A succinct statement emphasizing the disagreement in detection among both fire data sources, as rightly indicated on line # 399 in the Conclusions section, will be helpful. There is the need to double-check and re-organize your references throughout the revised manuscript. For instance, the following references were not cited in the text but have been included in the list of literature cited in References section - reference #2, # 6, #8 etc. Line #46: [1,1]? Line #57: [1,15]; Meanwhile, … change “;” to full stop.Author Response
Please see the attachment. Thanks.
